# Phenylethyl Isothiocyanate Extracted from Watercress By-Products with Aqueous Micellar Systems: Development and Optimisation

**DOI:** 10.3390/antiox9080698

**Published:** 2020-08-03

**Authors:** Ezequiel R. Coscueta, Celso A. Reis, Manuela Pintado

**Affiliations:** 1CBQF—Centro de Biotecnologia e Química Fina—Laboratório Associado, Escola Superior de Biotecnologia, Universidade Católica Portuguesa, Rua Diogo Botelho 1327, 4169-005 Porto, Portugal; 2i3S—Instituto de Investigação e Inovação em Saúde, Universidade do Porto, 4200-135 Porto, Portugal; celsor@ipatimup.pt; 3Institute of Molecular Pathology and Immunology of University of Porto, Ipatimup, 4200-135 Porto, Portugal; 4Medical Faculty, University of Porto, Al. Prof. Hernâni Monteiro, 4200-319 Porto, Portugal

**Keywords:** phenylethyl isothiocyanate (PEITC), aqueous micellar systems, aliphatic alcohol ethoxylates, watercress by-products, response surface methodology, optimisation, antioxidant activity, Gastrocure

## Abstract

Phenylethyl isothiocyanate (PEITC) was reported as a useful antioxidant, anti-inflammatory, and chemopreventive agent. Due to technological and stability issues, it is necessary to be able to extract PEITC from its natural matrix (watercress) through sustainable and scalable methodologies. In this article, we explored, for the first time, the extractive capacity of aqueous micellar systems (AMSs) of two non-ionic surfactants. For this, we compared the AMSs with conventional organic solvents. Furthermore, we developed and optimised a new integral PEITC production and extraction process by a multifactorial experimental design. Finally, we analysed the antioxidant capacity by the oxygen radical absorbance capacity (ORAC) and ABTS methods. As results, the AMSs were able to extract PEITC at the same level as the tested conventional solvents. In addition, we optimised by response surface methodology the integrated process (2.0% m/m, 25.0 °C, pH 9.0), which was equally effective (ca. 2900 µg PEITC/g watercress), regardless of the surfactant used. The optimal extracts showed greater antioxidant capacity than pure PEITC, due to other antioxidant compounds extracted in the process. In conclusion, by the present work, we developed an innovative cost-effective and low environmental impact process for obtaining PEITC extracts from watercress by-products.

## 1. Introduction

Some cruciferous plants are classed as “superfood”, because they may have beneficial effects in the treatment and prevention of several health disorders. This property is ascribed to their complex nutritional and phytochemical profile, which includes high concentrations of carotenoids, flavonoids, phenolic acids, and isothiocyanates [1]. Isothiocyanates are one of the hydrolysis products of glucosinolates (a class of sulphured secondary metabolites from the botanical order *Brassicales*) by the enzyme myrosinase [2,3]. In the intact plant, myrosinase is stored separately from glucosinolates; when the plant tissue is damaged (by chopping or chewing), the enzyme contacts with glucosinolates and catalyses the lysis [3,4]: the reaction produces indoles, nitriles, thiocyanates, or isothiocyanates. Glucosinolates are water-soluble and stable compounds, but isothiocyanates are hydrophobic and very reactive compounds.

Among the aforementioned superfoods, watercress (*Nasturtium officinale*—WC) is the richest known source of the gluconasturtiin (phenylethyl glucosinolate) derived phenylethyl isothiocyanate (PEITC) [5]. PEITC—one of the most interesting isothiocyanates—is a useful antioxidant, antimicrobial, and cancer chemo-preventive agent [3,4,6]. Unfortunately, direct consumption of WC does not ensure the amount and bioaccessibility of PEITC, since, from a biological point of view, the formation of PEITC is easily affected by various factors such as temperature, pH, and presence of additives [3,7,8]. Besides that, from a food-technological point of view, myrosinase is susceptible to heat and can be inactivated during the cooking process, inhibiting the hydrolysis of glucosinolates [7].

Therefore, the production and extraction of PEITC from WC discards may represent an opportunity to extract it in a stable and bioactive form taking advantage of an important by-product, generating great value. In that sense, the reported work is scarce and, even more, the existent studies are not using a sustainable approach [9,10,11]. The applied methodologies, in general, are analytical and use polluting organic solvents that are unfeasible at the industrial level. The few works with sustainable methodologies apply more complex techniques and expensive, and not easily scalable [9,12,13]: microwave-assisted ethanol extraction, supercritical fluids.

In recent decades, a wide range of new non-toxic, non-flammable, and biodegradable solvents have been evaluated to develop environmentally friendly and sustainable extraction methods [14]. Among them, certain surfactants have the mentioned properties, which represent an economic alternative to expensive and dangerous organic solvents. Non-ionic surfactants represent an important class of amphiphiles, which find wide applications in pharmaceutical and industrial formulations, and are widely used in the extraction process and purification of compounds of biological origin [15,16,17,18]. As indicated by Katsoyannos et al. [18], micellar systems using non-toxic surfactants (non-ionic, without branched aliphatic chains or aromatic moieties) are suitable for the isolation of natural antioxidants, which can then be used in dietary applications. Among non-ionic surfactants, polyethoxylates are the most numerous and technically interesting [15], standing out for their widespread use Triton (X-100 and X-114), Brij (−30, −56 and −97), Genapol (X-080), and to a lesser extent, the Tergitols (15-S-X), among others [19,20,21,22,23]. Genapol X-080 is a structured non-ionic surfactant of the ethoxylated primary aliphatic alcohols type, with a hydrophobic chain of 12 alkyls and 8 oxyethylene groups (OEs) in its hydrophilic region. For their part, the non-ionic surfactants Tergitol 15-S-7 are a mixture of ethoxylated secondary aliphatic alcohols with 11–15 carbons in the hydrophobic alkyl chain and with an average number of OEs of 7.3 as hydrophilic head. Both Genapol X-080 and Tergitol 15-S-7 are biodegradable and non-toxic. Furthermore, they have the quality of being transparent in the UV 240–280 nm region, facilitating the monitoring of purification processes of aromatic molecules or with conjugated double bonds that adsorb in that spectral zone. The micelles present in colloidal dispersions (aqueous micellar systems—AMSs) of these surfactants can interact with hydrophilic or hydrophobic molecules through different types of interactions, creating conditions for separate purposes [24]. Previously, we have developed and optimised extractive processes with AMSs to recover low polar compounds, obtaining high yields [19,25,26]. Currently, there is no study on AMSs for the extraction of PEITC, or any isothiocyanate, which opens the doors for an interesting analysis.

In this framework, we carried out a study to apply AMSs to produce extracts enriched with PEITC, developing a profitable process with low environmental impact. As a road map, we pose the following questions: Are Genapol X-080 and Tergitol 15-S-7 AMSs suitable for PEITC extractants, comparable to currently used solvents? What are the variables that most affect the extractive process of PEITC with AMSs? What are the optimal conditions that maximize PEITC extraction?

## 2. Materials and Methods

### 2.1. Materials

The surfactants Genapol X-080 (GX080) and Tergitol 15-S-7 (Tg7) were supplied from Sigma-Aldrich (St. Louis, MO, USA). Methanol, acetonitrile (ACN), chloroform (CF), and n-hexane were of HPLC grade. All the reagents were used as received without further purification. PEITC standard (Santa Cruz Biotechnology, Inc., Dallas, TX, USA) were dissolved in pure methanol until obtaining 22–660 µg/mL. All the other reagents were of analytical grade and used without further purification. Ultrapure water was used to prepare all the solutions.

Fresh WC by-products (discards from the washing process and selection of plant material for direct consumption) was kindly supplied by the food processing company Vitacress Portugal SA (Odemira, Portugal). When received, WC was dehydrated in a dryer at 50 °C for 24 h [11] and then pulverized and stored in a hermetic container.

### 2.2. Extraction Procedures

An exploratory analysis of the efficacy of different solvents to extract PEITC was performed; 1.00 g of the pulverized WC was suspended in 15 mL of 25 mM phosphate buffer pH 7.0 for 2 h at 37 °C to allow the hydrolysis of the glucosinolates [10]. Then, 0.15 g of surfactant (Genapol X-080, Tergitol 15-S-7), or 5 mL of n-hexane, or 5 mL of ACN/CF 10/7 mixture was added. The heterogeneous matrix–solvent systems were mixed vigorously for 10 min at a constant temperature (37.0 °C) and then centrifuged in Eppendorf tubes at 5000 rpm for 10 min. The whole supernatant (in case of SMAs) or the organic phase (in case of aqueous/organic solvent systems) were separated for analytical evaluation.

### 2.3. Optimisation of PEITC Extraction with AMSs

An integrative process to produce and extract PEITC was developed and optimised. To do this, an experimental strategy was implemented to reduce the number of experiments necessary to optimise the process, as well as to establish the most influential factors. For this purpose, a Box-Behnken design was selected, which is a type of 3-level response surface design that includes a subset of runs of a full factorial at three levels. The factors evaluated were surfactant concentration (X_A_), temperature (X_B_), and pH value (X_C_) and the selected response variable (Y) was the PEITC content, as above. The design resulted in an arrangement of 15 treatments, which was executed in triplicate (a total of 45 runs) on successive days. The levels of the factors, coded as −1 (low), 0 (central point), and +1 (high), are shown in Table 1. To carry out the experiments, we proceeded as described above, with the difference that instead of incubating only in the buffer, it was incubated in AMS. The AMSs for the different pH values were prepared in citrate buffer for pH 5.0, phosphate buffer for pH 7.0, and tris buffer for pH 9.0. After incubation, the extracts were filtered with 0.45-micron filters and stored at −20 °C for further analysis.

### 2.4. PEIT Chromatographic Analysis

PEITC extracts were diluted and analysed by high-performance liquid chromatography (HPLC-DAD). The separation was carried out by applying the operating conditions described by Fusari, Ramirez, and Camargo [10], with slight modifications. Briefly, the operating conditions were as follows: gradient elution starts at 50% mobile phase A (ultra-pure water + 0.1% formic acid) and 50% mobile phase B (methanol + 0.1% formic acid), with a phase B increment up to 80% for 20 min, remaining isocratic for 10 min, and returning to 50% of phase B the next 5 min, at a continuous flow of 0.80 mL/min. The procedure was performed in a reversed-phase column coupled with a guard column containing the same stationary phase (COSMOSIL 5C18-AR-II Packed Column —4.6 mm I.D. × 250 mm). The separation analysis was performed using a Waters e2695 separation module system interfaced with a photodiode array UV/Vis detector (PDA 190–600 nm). The identification of the WC PEITC peak was made by comparing the retention time and the absorption spectrum with the corresponding to a commercial standard. The quantification was carried out by integrating the peak area with external standardization.

### 2.5. Antioxidant Activity

Oxygen radical absorbance capacity (ORAC) assay was performed following the methodology used by Coscueta et al. [27], with some modifications. The reaction was carried out in 75 mM phosphate buffer (pH 7.4), and the final reaction mixture was 200 µL. Antioxidant (20 µL) and fluorescein (120 µL; 70 nM, final concentration in well) solutions were placed in the well of the microplate. A blank (fluorescein + AAPH) using phosphate buffer instead of the antioxidant solution and eight calibration solutions using Trolox (1–8 µM, final concentration in well) as antioxidant were also carried out in each assay. The mixture was preincubated for 10 min at 37 °C. AAPH solution (60 µL; 12 mM, final concentration in well) was added rapidly. The microplate was immediately placed in the reader and the fluorescence recorded at intervals of 1 min during 80 min. This assay was performed with a multidetection plate reader (Synergy H1, Vermont, USA) controlled by the Gen5 Biotek software version 3.04. The excitation wavelength was set at 485 nm and the emission wavelength at 528 nm. The microplate was automatically shaken before each reading. Black polystyrene 96-well microplates (Nunc, Denmark) were used. AAPH and Trolox solutions were prepared daily and fluorescein was diluted from a stock solution (1.17 mM) in the same phosphate buffer. Antioxidant curves (fluorescence versus time) were first normalized to the curve of the blank corresponding to the same assay by multiplying original data by the factor fluorescence_blank,t=0_/fluorescence_control,t=0_. From the normalized curves, the area under the fluorescence decay curve (AUC) was calculated according to the trapezoidal method. The final AUC values were calculated by subtracting the AUC of the blank from all the results. Regression equations between net AUC and antioxidant concentration were calculated. Final ORAC values were expressed as μmol TE (Trolox equivalent)/mg PEITC.

ABTS scavenging assay was performed in a 96-well microplate, following the method described by Gonçalves et al. [28] with some modifications. Briefly, ABTS radical cation (ABTS^•+^) was produced from the reaction of 7 mM 2,20-azinobis (3-ethylbenzothiazoline-6-sulfonic acid) diammonium salt and 2.45 mM potassium persulfate (both from Sigma-Aldrich, St. Louis, MO, USA) after incubation at room temperature in the dark for 16 h. For ABTS^•+^ working solution (freshly prepared), the ABTS^•+^ was filtered with a 0.45 µm syringe filter and diluted with the solvent to an absorbance of 0.70 ± 0.02 at 734 nm for the control (20 µL solvent + 180 µL ABTS^•+^ working solution). To 20 µL of sample or Trolox (25–175 μM) or solvent was added 180 µL of ABTS^•+^ working solution. The mixture was incubated for 5 min at 30 °C, and the absorbance at 734 nm was measured with a multidetection plate reader (Synergy H1, Vermont, USA) controlled by the Gen5 Biotek software version 3.04. The scavenging activity was expressed as % reduction in absorbance for the control. Regression equations between net ABTS scavenging and Trolox concentration were calculated and the results expressed as μmol TE (Trolox equivalent)/mg PEITC.

### 2.6. Statistical Analysis

Each experiment was performed in triplicate and the results were expressed as the mean values with standard deviations (SD). The major statistical analysis was carried out with the aid of RStudio V 1.2.1335.

Comparison of extraction efficiency: The mean PEITC values were analysed statistically by one-way analysis of variance (ANOVA) followed by the Tukey’s posthoc test [29,30]. Separation of means was conducted by the least significant difference at the 5% probability level.

Optimisation: The response values (Y), i.e., the PEITC in the supernatants from each treatment, were fitted to the following polynomial quadratic model:(1)$$Y=\beta_{0}+\beta_{A}X_{A}+\beta_{B}X_{B}+\beta_{C}X_{C}+\beta_{A,A}X_{A}^{2}+\beta_{A,B}X_{A}X_{B}+\beta_{A,C}X_{A}X_{C}+\beta_{B,B}X_{B}^{2}+\beta_{B,C}X_{B}X_{C}+\beta_{C,C}X_{C}^{2}+\varepsilon$$
where *X_A_* and *X_B_* are the coded levels of the independent variables mentioned above; β_0_, β_i_, β_i,I_, and β_i,j_ are the regression coefficients for the independent term, the linear, quadratic, and binary interaction effects, respectively; and ε, the residual error [25,31]. The surface and contour graphs of the responses were generated from this model.

## 3. Results

### 3.1. Liquid-Liquid Extraction: Comparison of Extractants

In the extraction of isothiocyanates, liquid-liquid extraction is generally applied. The highly water-soluble glucosinolates are hydrolysed in the aqueous medium by myrosinase, producing the isothiocyanates. The highly hydrophobic isothiocyanates were extracted from the aqueous medium by an immiscible solvent (phase separation) or by decreasing polarity with a less polar miscible solvent. We tested a set of different solvent systems as PEITC extractants; we compared AMSs of two aliphatic alcohol ethoxylates (Genapol X-080 and Tergitol 15-S-7) with conventional organic solvents (Table 2). Systems with Tergitol 15-S-7 and ACN/CF did not differ significantly from n-hexane, while the AMS of Genapol X-080 extracted slightly less (Table 2).

### 3.2. Optimised PEITC Extraction

After corroborating the extractive efficacy of the studied AMSs, we decided to develop an integrated and optimal process for the production and extraction of PEITC. To do this, we applied the conditions to produce autolysis directly in the AMS. After the incubation time, it was only necessary to separate the remaining solid material by filtration.

According to literature, many variables affect the extraction process of a given substance from a solid matrix, i.e., temperature, extraction time, pH, the particle size of the solid, stirring rate, and extractant concentration [32]. In this case, we decided simultaneously to optimise autolysis and extraction based on the surfactant concentration, incubation temperature, and pH. The time remained constant, since it has already been reported as ideal for 2 h [10]. Considering this, we selected an experimental design of the Box-Behnken type (described in Section 2.3). We selected the maximum and minimum values for the evaluated factors according to our knowledge of the systems.

Table 3 shows the design matrix and the multiple regression results obtained for the 15 randomized treatments, both for Genapol X-080 and for Tergitol 15-S-7. Figure 1 shows that the effects were equally significant among the surfactants. In this sense, the linear effects of pH and surfactant concentration were significant and proportional to the PEITC obtained. On the other hand, the linear effect of temperature was the third most significant effect, but with an inverse contribution to the PEITC content. Furthermore, the linear interaction between the factors was significant for both surfactants.

Then, we recalculated the models, by eliminating the non-significant effect ($X_{A}^{2}$), with the criterion of maximizing the R^2^_adj_, fitting again from the multiple regression of the data. Table 4 shows the parameters of the final models, including the obtained regression statistics: lack of fit, R^2^, R^2^_adj_, and RSD. The new models adequately explain the variation in the response, with a maximum R^2^_adj_. Figure 2 shows the response surfaces of the final models, expressing the response as a function of the factors considered for each surfactant.

Looking at the response surfaces, we saw the increments in surfactant concentration and pH led to increments of the response for both surfactants. However, the temperature, with a more marked curvature for Tergitol 15-S-7, maximizes the response to intermediate values, but close to the lower level (−1), 25 °C. The fitted models predicted maximum PEITC values of 3199 ± 338 µg PEITC/g WC DB and 3306 ± 312 µg PEITC/g WC DB, respectively, for Genapol X-080 and Tergitol 15-S-7. The optimal condition to apply in the process with the AMSs would be 2.0% m/m at 25.0 °C and a pH value of 9.0 for both surfactants. As shown by the extractant comparison, in the optimal conditions of the process developed for the AMSs, the predicted PEITC values are not different between Genapol X-080 and Tergitol 15-S-7, similarly to what we reported for the extraction of soy isoflavones [25]. Then, we applied the said optimal condition to validate both models. Thus, we obtained PEITC values of 2887 µg PEITC/g WC DB with Genapol X-080 and 2971 µg PEITC/g WC DB with Tergitol 15-S-7, values that, although they were close to the lower limit of the predicted confidence interval, were still within the prediction.

### 3.3. Antioxidant Activity

We analysed the antioxidant activity of the optimised extracts, using the ORAC and ABTS methodologies. Table 5 shows the results obtained, indicating a clear difference between the antioxidant activity of the extracts with the AMSs and the pure PEITC.

## 4. Discussion

The use of conventional extraction with n-hexane has led to obtaining an amount of PEITC (Table 2) almost half of that reported by Rodrigues et al. (3346 µg PEITC/g WC DB) under similar conditions [12]. Farhana et al., reported higher levels, but not distant values (ca. 2300 µg/g WC DB), using dichloromethane as an extractant under similar conditions [7]. The production of isothiocyanates can vary, depending on the conditions to which the plant is subjected, more specifically, the temperature and pH value to which the reaction occurs. Besides that, the production also depends on the plant species and age, as well as other factors such as place of cultivation, climatic conditions, storage, and processing [4]. Considering this, more than in extraction, the difference lies in the production of PEITC itself. The use of the ACN/CF mixture, as an extractant in dispersive liquid-liquid microextraction (DLLME) technique, was developed and validated by Fusari et al. [10] to determine the content of isothiocyanates in Brassicaceae vegetables. Taking these two methodologies as references, we can assert the efficacy of the analysed AMSs as PEITC extractants.

Nevertheless, we asked the question: Why are these AMSs so effective as organic solvents in extracting a hydrophobic molecule like PEITC? Genapol X-080 and Tergitol 15-S-7 have high effectiveness in interacting with low polar molecules in plant matrices. This appears to be closely related to the amphiphilic character of these surfactants and its ability to form micelles that can interact with hydrophobic groups of PEITC molecule. As previously discussed, it should be noticed that Genapol X-080 and Tergitol 15-S-7 have critical micellar concentrations (CMCs) of 4.6 10^−3^ and 3.9 10^−3^% m/m, respectively [25]. The concentrations applied in our study were higher than those CMCs. Then, by working on a concentration above the CMC, we provide a favourable environment (micelle) for the dispersion of PEITC in the aqueous solution.

Concerning the factors that affect the extractive process, Farhana, et al., with a one-factor approach, also observed similar effects of pH and temperature on PEITC production [7]. In that study, they reported that myrosinase activity did not vary between 25 °C and 45 °C, decreasing slightly towards 65 °C. Regarding the pH, the enzymatic activity remained stable between pH 7.0 and 9.0, while, below 7.0, it decreased slightly. However, to produce PEITC, they found more marked behaviours than on enzymatic activity. They established that, at acidic pH, nitrile production is more favoured and therefore PEITC production decreases. However, from their univariate approach, they failed to reliably explain the variability that was not explained by the enzymatic activity of myrosinase. The authors concluded that the higher the activity of the enzyme, the greater the production of PEITC, this being optimal at mild temperatures (ca. 25 °C) and a pH value between neutral to slightly alkaline. Furthermore, they obtained a maximum PEITC production of 5611 µg PEITC/g WC DB in conditions like those predicted as optimal by our polynomial models.

At this point, it is important to mention the appropriateness of using solid statistical tools in the optimisation process. Biological matrices are complex systems strongly affected by different variables and their interactions, making it practically unfeasible to predict accurate behaviours based on the classic one-factor method (one variable at a time). Both multifactorial models indicate a maximisation towards one of the vertices of the design, leaving us the mystery of what could happen in conditions beyond that vertex, that is, at factor levels beyond those analysed. However, within the tested conditions, we can talk about an interesting scenario: these predictions of a unified condition, with similar results, provide the desired scenario of flexible technology, applicable more generally and not restricted to a particular surfactant.

Regarding antioxidant activity, the differences between the pure PEITC and the extracts may be due to the other phytochemicals that the AMSs were able to extract, such as phenolic compounds [19,25]. Rodrigues et al. carried out pressurized fluid extraction in watercress and reported a relationship between the extractive increase of phenolic compounds and the antioxidant activity (ORAC) independent of the PEITC content [12]. Now, this does not mean that PEITC does not have antioxidant capacity. PEITC has been reported as an antioxidant compound, but not by direct action on radicals, but by reducing the reactive oxygen species (ROS) load, being an important inducer/enhancer of antioxidant/detoxification enzymes [33,34,35]. Besides, PEITC may act simultaneously as an oxidant, since it has a high capacity to generate ROS and induce oxidative damage in tumour cells, which makes PEITC “selective” for this cell type [33,36,37]. Thus, the extracts produced have the advantage of providing an antioxidant action through different routes, synergistically combining the properties of PEITC and phenolic compounds.

## 5. Conclusions

Although there has been a growing interest in isothiocyanates in recent years, the information available is still scarce. There is practically no solid base of work with a technological focus and still less for the PEITC. Therefore, considering the potential of this compound and the technological relevance it may have soon, an exploratory study of scalable green methodologies for its production and extraction is of high impact.

In the present study, we report the development of a methodology to extract PEITC by applying cost-effective and low environmental impact systems, reaching the extractive capacity of classical organic solvents. Thus, our exploratory and optimisation work—with robust statistical tools—allows us to contribute with an innovative precedent in the state of the art.

The flexibility of selection of different surfactants (biodegradable, non-flammable, and non-toxic) for PEITC extraction with similar results, as well as the possibility of substituting organic solvents in large-scale extraction procedures, make these AMSs suitable for industrial production of PEITC from WC.

As a final consideration, the extracts obtained resulted in an interesting antioxidant product, which may synergistically combine different mechanisms of direct and cellular action against harmful oxidative states. However, this must be studied in greater depth and validated by other methodologies.

## Figures and Tables

**Figure 1 antioxidants-09-00698-f001:**
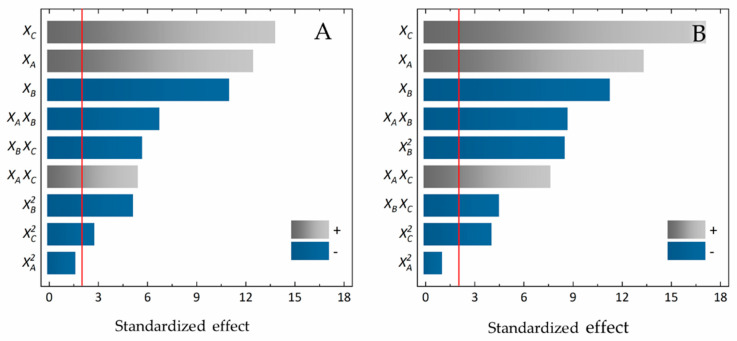
Analysis of the effects of the Box-Behnken factorial design. Pareto charts with standardized effects of three experimental factors, in decreasing order of importance (in absolute value) for the phenylethyl isothiocyanate (PEITC) content response. Genapol X-080 (**A**) and Tergitol 15-S-7 (**B**) aqueous micellar systems (AMSs). The vertical red lines represent the threshold of significance (*p* = 0.05) for 32 degrees of freedom.

**Figure 2 antioxidants-09-00698-f002:**
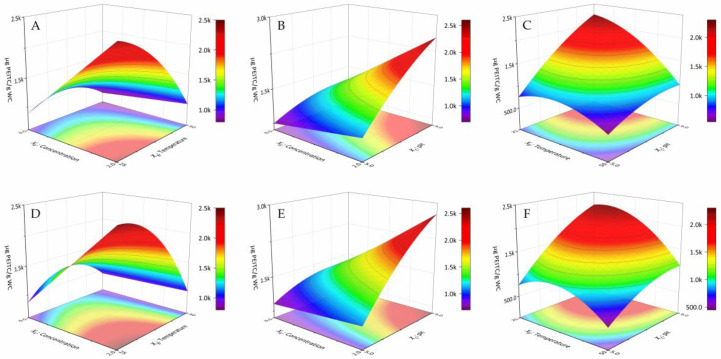
Response surface models for the response estimated based on two experimental factors keeping the third factor at its central level. Charts for Genapol X-080 (**A**–**C**) and Tergitol 15-S-7 (**D**–**F**).

**Table 1 antioxidants-09-00698-t001:** Levels for 3 experimental factors.

Factors	Levels		
	−1	0	+1
X_A_ ^1^	0.50	1.25	2.00
X_B_ ^2^	25.0	37.5	50.0
X_C_ ^3^	5.0	7.0	9.0

^1^ Surfactant concentration expressed as % m/m, ^2^ temperature expressed as °C, and ^3^ pH value.

**Table 2 antioxidants-09-00698-t002:** Comparison of extractants.

	Extractant	Conc. (µg/g WC DB)
WC extracts	Genapol X-080	1168 ± 187 ^a^
	Tergitol 15-S-7	1287 ± 83 ^a,b^
	n-hexane	1682 ± 156 ^b^
	ACN/CF	1270 ± 169 ^a,b^

^a,b^ Values of the same column that share superscript do not present a significant difference, given the analysis of their variances (ANOVA) with the Tukey test for a level of significance of 0.05.

**Table 3 antioxidants-09-00698-t003:** Box-Behnken factorial design for 3 factors and one response, for Genapol X-080 and Tergitol 15-S-7 AMSs.

Run	Factors			Response ^†^	
	XA 1	XB 2	XC 3	Y (µg PEITC/g WC DB)	
				Genapol X-080	Tergitol 15-S-7
1	1.25	50.0	5.0	641 ± 102	590 ± 68
2	0.50	37.5	5.0	654 ± 69	798 ± 86
3	2.00	37.5	9.0	2552 ± 247	2842 ± 229
4	1.25	37.5	7.0	1450 ± 175	1669 ± 173
5	0.50	37.5	9.0	1168 ± 122	1346 ± 97
6	1.25	50.0	9.0	939 ± 79	1090 ± 63
7	1.25	25.0	9.0	2235 ± 149	2102 ± 156
8	1.25	37.5	7.0	1640 ± 181	1760 ± 163
9	0.50	25.0	7.0	852 ± 70	891 ± 65
10	2.00	25.0	7.0	2346 ± 169	2441 ± 185
11	2.00	50.0	7.0	940 ± 111	932 ± 96
12	2.00	37.5	5.0	988 ± 144	971 ± 85
13	1.25	25.0	5.0	838 ± 68	824 ± 84
14	0.50	50.0	7.0	748 ± 69	887 ± 70
15	1.25	37.5	7.0	1593 ± 142	1722 ± 103

^†^ Values expressed as mean ± SD of three replicates.

**Table 4 antioxidants-09-00698-t004:** Better estimates of the coefficients of each term in the model and the corresponding statistics.

Coefficient	Estimated Coefficient Values	
	Genapol X-080	Tergitol 15-S-7
$\beta_{0}$	−6605.12	−7396.17
$\beta_{A}$	645.722	504.556
$\beta_{B}$	211.279	259.502
$\beta_{C}$	895.638	903.736
$\beta_{A,B}$	−34.7289	−40.1244
$\beta_{A,C}$	174.833	220.611
$\beta_{B,B}$	−1.61313	−2.432
$\beta_{B,C}$	−10.99	−7.79
$\beta_{C,C}$	−33.3045	−44.6458
**Statistics**		
Lack of fit ^1^	0.351	0.082
R^2^	0.943	0.953
R^2^_adj_	0.930	0.943
RSD	167	150

^1^ Lack of fit test expressed in its test value (*P*).

**Table 5 antioxidants-09-00698-t005:** Antioxidant activity by oxygen radical absorbance capacity (ORAC) and ABTS.

Sample	ORAC (μmol TE/mg PEITC)	ABTS (μmol TE/mg PEITC)
PEITC	1.9 ± 0.7	1.1 ± 0.2
AMS1 ^1^	134.9 ± 4.0 ^a^	86.4 ± 3.8 ^a^
AMS2 ^2^	123.3 ± 5.1 ^a^	81.2 ± 2.6 ^a^

^1^ Extract in optimised conditions for Genapol X-080 AMS. ^2^ Extract in optimised conditions for Tergitol 15-S-7 AMS. ^a^ Values of the same column that share superscript does not present a significant difference, given the analysis of their variances (ANOVA) with the Tukey test for a level of significance of 0.05.

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
