# Peer review of "Phenylethyl Isothiocyanate Extracted from Watercress By-Products with Aqueous Micellar Systems: Development and Optimisation"

_antioxidants, 2020, doi:10.3390/antiox9080698_

Round 1

Reviewer 1 Report

This work reports a more biocompatible and sustainable approach for the extraction of phenylethyl isothiocyanate from watercress with interesting results. The manuscript is well prepared with every section being properly presented and explained. I have one question though, why haven't the authors used 5 mL of each surfactant while doing the screening alongside the different organic solvents? Both surfactants are liquids...

Author Response

We greatly appreciate the reviewer's comment. Concerning the question posed, it is not the same to think about the volumes of organic solvents as of surfactants. We work with micellar systems about the final concentration of the surfactant, so a volume of 5 mL of surfactant in a final volume of AMS of 20 mL is very high (almost 25 % w/w), an unfeasible concentration that produces a highly viscous system. When we talk about extractions with aqueous micellar systems of this type, in practice we never work with more than 5% w/w.

Reviewer 2 Report

The topic of the research presented in the manuscript is novel and interesting, describing the optimisation of the aqueous micellar system extraction procedure for the enhanced recovery of PEITC from watercress.

Please find below a few suggestions that may improve the quality of the manuscript:

In the Introduction section please present the possible applications of the extracts obtained using described AMS (food industry, cosmetic industry?)

The Materials and methods section would be more clear is the PEITC determination and the antioxidant activity analysis would be described as separate subsections.

The data presented in Table 5 shows a great difference in the antioxidant activity between pure PEITC, AMS1 and AMS2 – the Authors mentioned in the discussion that this high antioxidant activity most likely results from the co-extraction of other bioactive phytochemicals such as phenolic compounds. For that reason the qualitative analysis of the content of AMS1 and AMS2 extracts would be very informative and would significantly increase the scientific value of this work. I would strongly encourage the Authors to include the data regarding the content of other compounds of AMS1 and AMS2 in the manuscript.

Author Response

The topic of the research presented in the manuscript is novel and interesting, describing the optimisation of the aqueous micellar system extraction procedure for the enhanced recovery of PEITC from watercress.

Please find below a few suggestions that may improve the quality of the manuscript:

We greatly appreciate the reviewer's suggestions. In the text, we will respond line by line, to particularly attend to each of the suggestions.

In the Introduction section please present the possible applications of the extracts obtained using described AMS (food industry, cosmetic industry?)

We found an enriching suggestion, which we tried to capture in the manuscript.

The Materials and methods section would be more clear is the PEITC determination and the antioxidant activity analysis would be described as separate subsections.

Following this suggestion, we separate these determinations into different sections, instead of including them in the same subsection of analytical determinations.

The data presented in Table 5 shows a great difference in the antioxidant activity between pure PEITC, AMS1 and AMS2 – the Authors mentioned in the discussion that this high antioxidant activity most likely results from the co-extraction of other bioactive phytochemicals such as phenolic compounds. For that reason, the qualitative analysis of the content of AMS1 and AMS2 extracts would be very informative and would significantly increase the scientific value of this work. I would strongly encourage the Authors to include the data regarding the content of other compounds of AMS1 and AMS2 in the manuscript.

We very much appreciate the recommendation and we consider it is important to be able to characterize the extracts in their entirety. However, that is beyond the objective of this work, where the idea focuses on the extractive process and on the ability to extract PEITC by the AMSs and not so much on the extract as a final product. In the sense of that interest, a deeper work will be carried out that characterizes the extracts compositionally and in a greater spectrum of bioactivities, not just antioxidants.

Reviewer 3 Report

Dear Ladies and Gentlemen,

I have just few comments regarding the text editing and grammar. Please check your comma placement and your filler sentences. They are sometimes unnecessary. 

kind regards 

Author Response

We greatly appreciate the reviewer's comment, suggestions that improve work communication are always welcome and necessary. Following this, we will reread and try to make the alterations that we consider pertinent.

Kind regards,

Round 2

Reviewer 2 Report

The Authors provided sufficient explanations to my previous comments and improved the manuscript according to my suggestions. Therefore I recommend this manuscript for publication in a present form.